# PLSR-Based Assessment of Soil Respiration Rate Changes under Aerated Irrigation in Relation to Soil Environmental Factors

**Jiapeng Cui** [1,2,3] **and Feng Tan** [2,*]

1    College of Engineering, Heilongjiang Bayi Agricultural University, Daqing 163000, China
2    College of Electrical and Information, Heilongjiang Bayi Agricultural University, Daqing 163000, China
3    Branch of Suihua, Heilongjiang Academy of Agricultural Mechanization Sciences, Suihua 152054, China
*    Correspondence: bayitf@byau.edu.cn

**Abstract:** To ameliorate soil oxygen deficiencies around subsurface drip irrigation (SDI) drippers, aerated irrigation (AI) was introduced to supply aerated water to the soil through venturi installed in the SDI pipeline. The objectives of this study were to investigate the effect of AI on the soil respiration rate and the mechanism of regulation. The Daejeon experiment included two treatments: AI and unaerated SDI as a control check (CK), and used the National Soil Quality Zhanjiang Observation and Experiment Station as a platform to carry out a 2-year (2020–2021) positioning experiment. The effects on the soil respiration rate, soil temperature, soil water content, oxygen content, soil bacterial biomass and root biomass of the two treatments were established. The oxygen content, soil bacterial biomass and root biomass regression equation, using the partial least squares regression analysis (PLSR) algorithm and structural equation modeling (SEM), screened out the influence of soil respiration under AI treatment as the main soil environmental factor and driving mechanism of rate change. The results showed that compared with the control CK, the soil respiration rate, soil oxygen content, root biomass and soil bacterial biomass were significantly enhanced under AI treatment, the soil water content had a decreasing trend, and there was no significant difference in the effect on soil temperature between the different treatments. The regression fitting results showed that the soil respiration rate under both treatments was negatively correlated with soil temperature using a quadratic polynomial correlation, linearly correlated with the soil oxygen content, positively correlated with root biomass and soil bacterial biomass using power function and positively correlated with the soil water content using a cubic polynomial correlation. The PLSR and SEM results demonstrated that aerated irrigation technology could drive the increase in the soil respiration rate by changing the soil oxygen content, root biomass and bacterial biomass.

**Keywords:** PLSR; aerated irrigation; maize; soil respiration; SEM; drive mechanism

## 1. Introduction

Soil respiration is a process in which soil microorganisms, plant roots and detritivores produce $CO_2$ while consuming organic matter due to metabolism, which mainly includes the heterotrophic respiration of soil microorganisms and animals and the autotrophic respiration of plant roots [1], and it is an important method for the mutual exchange of soil and atmospheric carbon pools. According to statistics, soil respiration exports from $83 \times 10^9$ to $108 \times 10^9$ t of carbon to the atmosphere annually, which is more than 10 times that of fossil fuel emissions [2]. Changes in soil respiration will inevitably affect atmospheric carbon concentrations, exacerbate global warming and thus endanger the living environment and the future development of human beings, and these changes are mainly influenced by a combination of multiple factors such as farm management practices, soil environmental factors, biological factors and ecosystem types [3,4].

Aerated irrigation technology is based on the underground drip irrigation piping system and uses ventilation devices such as air pumps or fans to directly ventilate the crop

inter-root area or venturi devices to mix air in the form of microbubbles into the irrigation water for percolation, thereby regulating the soil water and gas content in the crop root zone to improve the crop inter-root soil environment [5–7]. Many studies have shown the potential and application of aerated irrigation in promoting crop growth and improving yields [8–17]. The beneficial effects of aerated irrigation on crops are directly attributed to the effective regulation of soil oxygen content and moisture by aerated irrigation. Soil moisture and gas have an important influence on crop growth and development and yield quality formation, and there is a coupling effect between the two. The regulation of aerated irrigation technology inevitably changes soil water content, oxygen content and microbial colonization, which in turn has an impact on soil respiration [18,19]. Studies have shown that soil aeration status and gas content directly affect the respiration of crop roots and the metabolic activity of microbial flora, which in turn affects the growth of crop roots, and these indicators are closely related to the soil respiration rate. When the oxygen content in the soil increases, it will significantly increase the microbial activity and root biomass in the soil, and the soil respiration rate will increase significantly [20–23]. At present, the response of soil respiration to different irrigation or fertilization measures and the mechanism have been reported, but the mechanism of the effect of aerated irrigation technology on the change in the soil respiration rate has mostly focused on the analysis of the effect of a single factor or two factors in the soil environment, such as the relationship between soil respiration and soil hydrothermal or soil oxygen content, and there is a lack of research on the comprehensive correlation analysis of soil respiration change with soil biotic and abiotic factors under aerated irrigation.

To address the above issues, maize was selected as the test material and conventional subsurface drip irrigation was used as the control experiment to study the differences and changes in the soil respiration rate under aerated irrigation technology. PLSR can effectively construct the most explanatory subspace regression model with multiple correlations of independent variables, significantly improving the accuracy and reliability of the model. In this study, the PLSR method was used to establish segmental regression equations for the soil respiration rate and soil environmental factors, to analyse the interaction between the soil respiration rate and soil temperature, moisture, oxygen content, bacteria and root biomass, and to reveal the main influences on the soil respiration rate under aerated irrigation based on the variable importance for projection (VIP) method.

## 2. Materials and Methods

### 2.1. Experimental Site

The treatment was conducted from 4 April 2020 to 15 November 2021 at the National Soil Quality Zhanjiang Observation Experiment Station of the Chinese Academy of Tropical Agricultural Sciences in Zhanjiang, Guangdong Province (109°31′ E, 21°35′ N), with an average annual sunshine duration of 2160 h, a frost-free period of 350 d and an average annual temperature of 23.2 °C, typical of a subtropical monsoon climate. The test soil was an in situ red loam in a maize field. Rainfall, temperature and other environmental factors were automatically obtained and recorded by a small weather station in the study area during the test period. Daily temperature and rainfall changes during the maize growing period from 2020 to 2021 are shown in Figure 1.

### 2.2. Experimental Design and Treatments

The treatment was set up with two irrigation methods: aerated irrigation (AI) and nonaerated irrigation (CK), and each treatment was replicated three times, with one replication for one plot and six plots in total. The treatment area was planted with a biannual maize cultivation pattern and the maize variety was "Huiyu Sweet No. 3". Before planting, standardized construction of farmland and the laying of subsurface irrigation piping system were completed in the trial area. Before planting, one underground drip irrigation belt (burial depth 20 cm, diameter 16 mm, flow rate 2.5 L/h, drip head spacing 20 cm) was laid in the middle of each plot, with the maize row spacing 60 cm and plant spacing 35 cm.

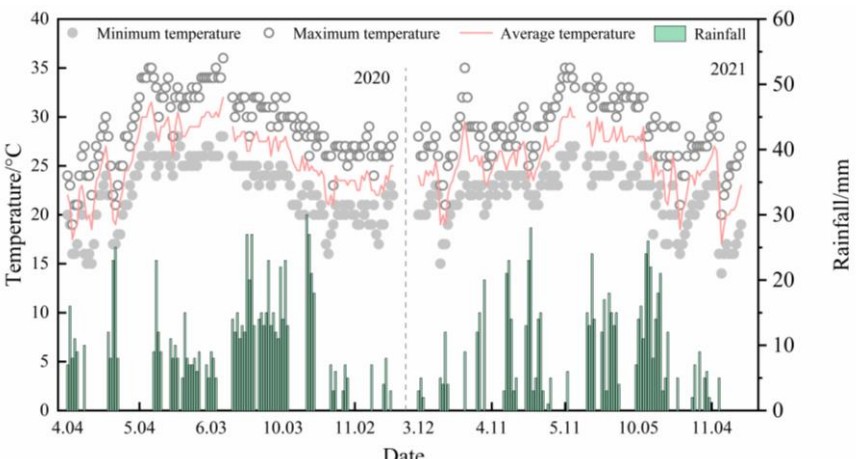

**Figure 1.** Average temperature (°C), maximum temperature (mm), minimum temperature (mm) and rainfall for each fertility period of maize in 2020–2021 at the National Experimental Station for Soil Quality Zhanjiang Observation.

The irrigation volume was controlled by ΦE601 standard evaporation dish (Equation (1)), and the irrigation interval was 1–3 days, and the irrigation time was between 08:00 and 12:00.

$$W = A \cdot E_P \cdot K_P \tag{1}$$

where $W$ is the amount of irrigation water per treatment, L; $A$ is the plot area controlled by a single drip head, 0.14 m$^2$ (0.35 m × 0.4 m); $E_p$ is the evaporation measured by a standard evaporation dish of ΦE601 type during the time interval between two irrigations, mm; $K_p$ is the crop-evaporation dish coefficient. The selection was based on the different fertility stages of maize, with 0.7 for seedling to elongation stage, 1.2 for elongation stage to tasseling and 0.6 for tasseling to maturity.

The entire reproductive cycle was aerated once every 2 d, with one additional aerating after each irrigation or rainfall, and the amount of aerating was calculated by Equation (2) [6], without considering the escape of gas from the soil in the treatment.

$$V = 1/1000SL(1 - \rho_b/\rho_s) \tag{2}$$

where $V$ is the volume per aeration, L; $S$ is the cross-sectional area of the monopoly, 1500 cm$^2$; $L$ is the length of the monopoly, m; $\rho_b$ is the soil capacitance, 1.2 g/cm$^3$; and $\rho_s$ is the soil density, 1.65 g/cm$^3$.

The aeration volume of each test plot was 744.75 L. During the test, the power and aeration volume of the roots blower were converted to the corresponding aeration time according to the roots fan nameplate, and the aeration volume was controlled by the aeration time without considering the escape of soil gas.

### 2.3. Measurement Indexes and Methods

Soil respiration rate: Soil respiration chamber measurements were carried out using the Li-8100A automatic soil carbon flux measurement system. Measurements were made during the 07:00–09:00 time period, and related studies have shown that the soil respiration rate measured during this time period is representative of the mean soil respiration rate for that day [20]. Measurements were taken every 10 days during the maize growth cycle, except for the start date and harvest date, and postponed in the case of heavy rainfall, and the mean values of different fertility periods of maize were taken in each treatment for statistical analysis.

Soil temperature, soil water content: The TZS-PHW-4G soil multifunctional parameter measuring instrument produced by Zhejiang Topunnong Technology Co., Ltd., (Hangzhou, Zhejiang Province, China) was used to track the soil temperature and moisture content

throughout the whole process, the technical parameters are shown in Table 1. The measurement depth was 20 cm from the soil, and the atmospheric temperature was recorded at the same time.

**Table 1.** TZS-PHW-4G Multifunctional Soil Parameter Tester.

| Technical Parameter | Test Range | Precision | Resolution |
|---|---|---|---|
| Temperature | −40–100 °C | ±0.5 °C | 0.1 °C |
| Water content | 0–100% | ≤3% | 0.1% |
| pH | 0–14 | ±0.5 | 0.1 |

Soil oxygen content: Using fiber-optic oxygen meter (fiber-optic oxygen meter firesting $O_2$) to determine the oxygen content at a distance of 30 cm from the soil surface, there is one oxygen meter in this treatment, which can be connected to two oxygen-sensitive probes at the same time, and the oxygen content changes in two plots can be measured at the same time, and the measurement date and time are consistent with the determination of soil respiration rate.

Soil bacterial biomass: Soil samples were collected from the cultivated layer by soil auger and five-point sampling method, and fresh soil samples were collected at 0–10, >10–20 and >20–30 cm soil layers and adequately by levels; each collection was repeated three times, and the soil bacterial count was determined using the plate counting method [9,13].

Root biomass: The root system was screened at a depth of 0–80 cm using a soil auger with the selected plants as the center, and the root–soil separation was conducted by the elution method; the samples were repeatedly sieved after soaking and stirring, and the roots were removed with forceps after root–soil separation and dried and weighed for roots.

*2.4. Data Processing and Analysis*

2.4.1. Partial Least Squares Regression Analysis

Partial least squares regression analysis (PLSR) is a multivariate statistical data analysis method proposed in 1968 as an integration and development of multiple linear regression, typical correlation analysis and principal component analysis. The PLSR method is a study of multiple independent variables to multiple dependent variables or a single dependent variable regression modelling method. It solves the problem of multiple correlation of independent variables in typical regression analysis, i.e., there is a high degree of correlation between independent variables. In this study, the PLSR approach of multiple independent variables to a single dependent variable was used. The main environmental factors affecting soil respiration rate under aerated irrigation technology were analysed using PLSR. In the PLSR model, variable importance for projection (VIP) is a multivariate screening method that describes the explanatory power of the independent variables to the dependent variable by combining the principal components of the independent variables of interest, and the independent variables are screened according to their explanatory power. The VIP value > 1 was considered to best explain the variation in soil respiration rate. The calculation procedure is as follows:

(1) Data standardization. The purpose of data normalization is to make the centre of gravity of the set of sampling points coincide with the origin of the coordinates, normalize the independent variable set X and the dependent variable to obtain a matrix in which the independent and dependent variables obey a normal distribution.

$$E_0 = \left[ \frac{x_{ij} - x_j}{S_j} \right]_{n \times p} \quad F_0 = \left[ \frac{y_{ij} - y}{S_y} \right]_{n \times 1} \quad i = 1, 2, \ldots\ldots, n; \; j = 1, 2, \ldots\ldots p \quad (3)$$

where $E_0$ is the normalized matrix of $x$, $F_0$ is a normalized matrix of $y$, $x_i$ and $S_j$ are the mean and standard deviation of $jth$, respectively, $y$ and $S_y$ are the mean and standard deviation of $y$, respectively.

(2) Find the value of the objective function for the optimization problem. First extract a component $t_1 = E_0\omega_1$, $\omega_1$ from the matrix $E_0$ that is the first principal axis of $E_0$ and $\|\omega_1\| = 1$. Then, a component $u_1 = F_0c_1$, $c_1$ is extracted from $F_0$ for the first principal axis of F and $\|u_1\| = 1$. $t_1$ and $u_1$ denote information about data changes in X and Y, $Var(t_1)$ and $V(u_1) \rightarrow Max$. Since correlation analysis requires that the independent variables have good explanatory power for the dependent variable, this requires $r(t_1, u_1) \rightarrow Max$. In the least partial squares regression analysis, the $t_1$ and $u_1$ covariances are required to be maximum and are calculated as in Equation (4):

$$Cov(t_1, u_1) = \sqrt{Var(t_1)Var(u_1)}r(t_1, u_1) \rightarrow Max \tag{4}$$

Thus, using the Lagrangian algorithm, one can turn to the problem of finding the maximum value of $\omega_1^T E_0^T F_0 c_1$ under the constraints of $\|\omega_1\| = 1$, $\|u_1\| = 1$:

$$s = \omega_1^T E_0^T F_0 c_1 - \lambda_1(\omega_1^T \omega_1 - 1) - \lambda_1(c_1^T c_1 - 1) \tag{5}$$

Then take the partial derivatives of $s$ about $\omega_1$, $c_1$, $\lambda_1$ and $\lambda_2$, respectively, and set them to 0.

$$\frac{\partial s}{\partial \omega_1} = E_0^T F_0 c_1 - 2\lambda_1 \omega_1 = 0 \tag{6}$$

$$\frac{\partial s}{\partial c_1} = F_0^T F_0 \omega_1 - 2\lambda_2 c_1 = 0 \tag{7}$$

$$\frac{\partial s}{\partial \lambda_1} = -\left(\omega_1^T \omega_1 - 1\right) = 0 \tag{8}$$

$$\frac{\partial s}{\partial \lambda_2} = -\left(c_1^T c_1 - 1\right) = 0 \tag{9}$$

The above four equations can be combined to obtain $2\lambda_1 = 2\lambda_2 = \omega_1^T E_0^T F_0 c_1$; let $\theta_1 = 2\lambda_1 = 2\lambda_2 = \omega_1^T E_0^T F_0 c_1$, $\theta_1$ is the objective value of the optimization problem.

(3) Substituting $\theta_1$ into the four equations above, the following equation is obtained:

$$\begin{cases} E_0^T F_0 F_0^T E_0 \omega_1 = \theta_1^2 \omega_1 \\ F_0^T E_0 E_0^T F_0 c_1 = \theta_1^2 c_1 \end{cases} \tag{10}$$

where, $\omega_1$ is the maximum eigenvalue of the matrix $E_0^T F_0 F_0^T E_0$ eigenvector and $c_1$ is the maximum eigenvalue of the matrix $c_1$ eigenvector. When $\omega_1$ and $c_1$ are derived, the principal components $t_1 = E_0\omega_1$ and $u_1 = F_0c_1$. The regression relationship between $E_0$, $E_0$ and $t_1$ is shown in Equation (11):

$$\begin{cases} E_0 = t_1 p_1^T + E_1 \\ F_0 = t_1 r_1^T + F_1 \end{cases} \tag{11}$$

(4) Derive the regression equation. If the accuracy of the regression equation for $y$ versus $t_1$ is not satisfied, the second component is extracted. At this point $E_0$ and $F_0$ are replaced by the residual matrices $E_1$ and $F_1$, and the principal axes $\omega_2$ and $c_2$, and the second principal components $t_2$ and $c_2$, are obtained again in the same way. $t_1, t_2 \ldots \ldots t_m$ always exists if the rank of F is m, giving Equation (12):

$$F_0 = t_1 r_1^T + t_2 r_2^T + \ldots \ldots + t_m r_m^T + F_m \tag{12}$$

since $t_1, t_2 \ldots \ldots t_m$ is a linear combination of $E_0$. Therefore, the above equation can be simplified to a regression equation of $x_j^* \sim E_0$ on $y^* \sim F_0$, giving Equation (13):

$$y^* = \alpha_1 x_1^* + \alpha_2 x_2^* + \ldots \ldots + \alpha_m x_m^* + F_m \tag{13}$$

Finally, normalization reduces the regression equation of $y^*1$ on $x_i^*$ to the regression equation of $X$ and $y$ in the inverse conversion process.

### 2.4.2. Structural Equation Modeling (SEM)

Compared with traditional statistical methods, SEM has many advantages: (1) SEM considers the correlations among multiple environmental factors; (2) SEM can clearly divide the effect into direct and indirect effects; and (3) SEM studies the influence of multiple factors on the performance of the model and evaluates the overall performance of the model [24].

It is known from existing studies that soil environmental factors can affect soil respiration rate in two ways: (1) by directly affecting soil respiration rate changes (e.g., soil temperature, moisture and oxygen content directly affect soil respiration rate), and (2) soil environmental factors affect soil respiration rate changes by affecting microbial respiration and crop growth. Therefore, in building the SEM model, we assumed that soil environmental factors directly affect soil respiration rate and indirectly affect soil respiration rate changes through soil biotic factors (microbial biomass and root biomass) [25].

### 2.4.3. Data Analysis

Statistical analysis of data was calculated by SPSS 22.0 software (IBM Corporation, Armonk, NY, USA) and SigmaPlot 10.0 (Systat Software, San Jose, CA, USA) using analysis of variance (ANOVA) to test the differences between the measured indicators between the two treatments. To quantify the relationship between soil respiration rate and soil water content, soil oxygen content, bacterial biomass and root biomass, ANOVA and regression analysis were performed by polynomial, exponential and linear models, and the best fit equation was screened by the coefficient $R^2$.

## 3. Results and Analysis

### 3.1. Characteristics of Changes in Soil Environmental Factors under Different Treatments

3.1.1. Soil Respiration Rate

There was some variability in the seasonal dynamics of the soil respiration rate (in terms of $CO_2$) under different treatments, and the measured points under AI treatment were all significantly higher than the control CK treatment ($p < 0.05$). Soil respiration under different treatments showed a general trend of increasing and then decreasing during the growing season of maize (Figure 2). The seasonal variation in soil respiration under AI was significantly greater than the control CK by 16.25–20.31% ($p < 0.05$) in the 2020–2021 treatment.

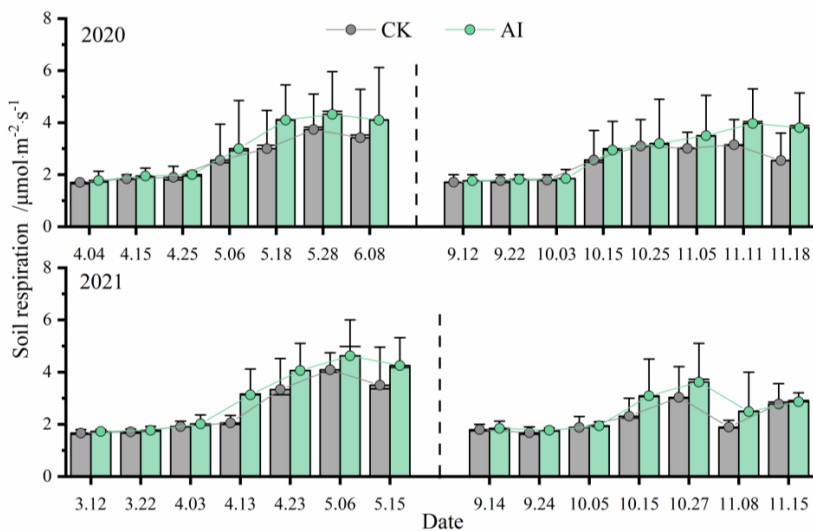

**Figure 2.** Dynamic change curve of soil respiration rate under different treatments (AI, aerated irrigation; CK, unaerated subsurface drip irrigation).

### 3.1.2. Soil Temperature

There was basically no significant difference ($p < 0.05$) between all measurement points of the soil temperature under different treatments, and only some of the measurement points of seasonal changes in the soil temperature under AI treatment were higher than those under CK treatment. Soil temperature was the main factor affecting the change in the soil respiration rate, which was also evident from the fact that the peak soil temperature corresponded significantly to the peak soil respiration rate (Figures 2 and 3).

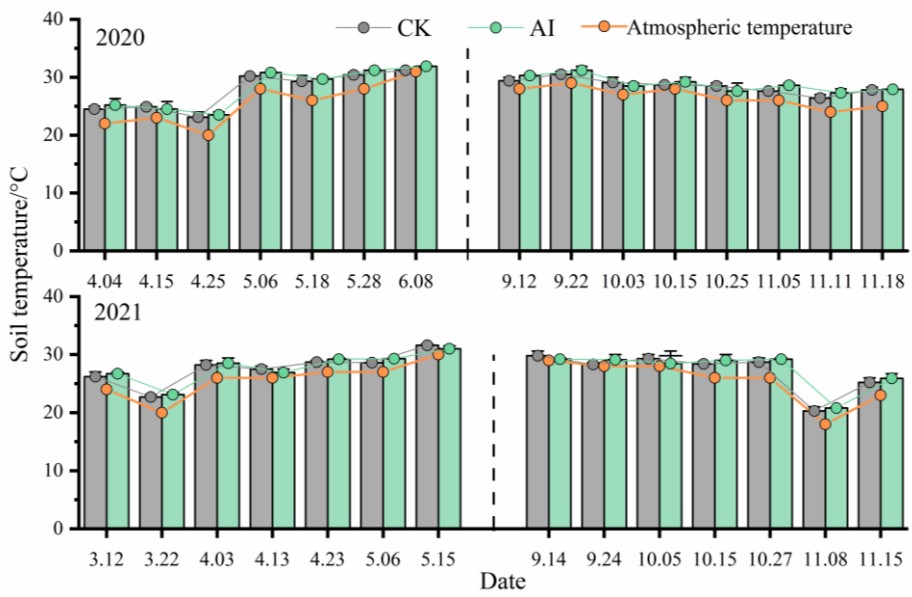

**Figure 3.** The dynamic change curve of soil temperature under different treatments (AI, aerated irrigation; CK, unaerated subsurface drip irrigation).

### 3.1.3. Soil Oxygen Content

In this study, all soil oxygen measurement points under AI treatment were significantly higher than the control treatment ($p < 0.05$) (Figure 4), and the soil oxygen content in AI was significantly higher than the control CK by 18.07–26.12% ($p < 0.05$) in the 2-year trial.

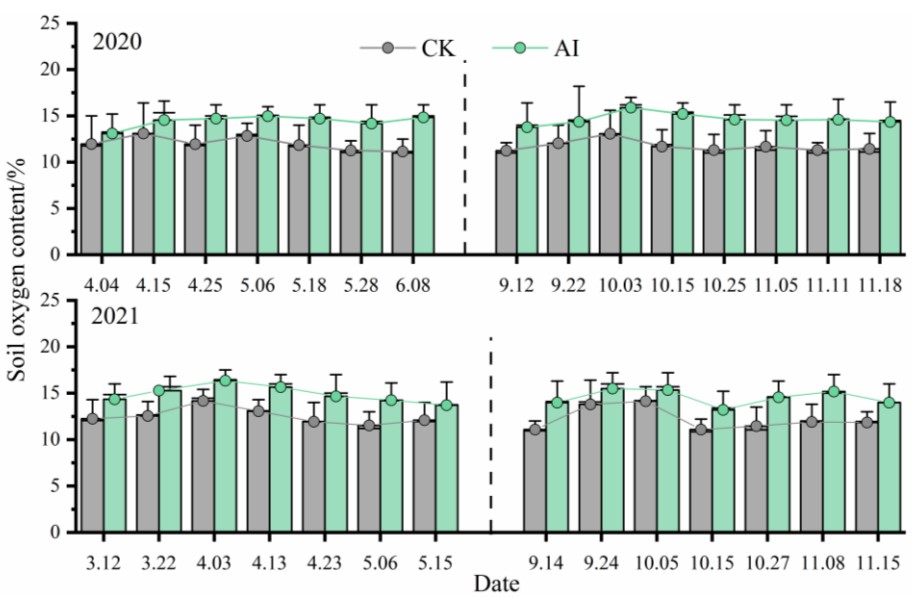

**Figure 4.** The dynamic change curve of soil oxygen content under different treatments (AI, aerated irrigation; CK, unaerated subsurface drip irrigation).

### 3.1.4. Soil Water Content

The soil water content variation in this study fluctuated, and the soil water content measurement point under AI treatment was basically lower than that of the control test (Figure 5); the soil water content under AI treatment decreased by 4.29–12.85% ($p < 0.05$) compared to the control CK in the 2-year test.

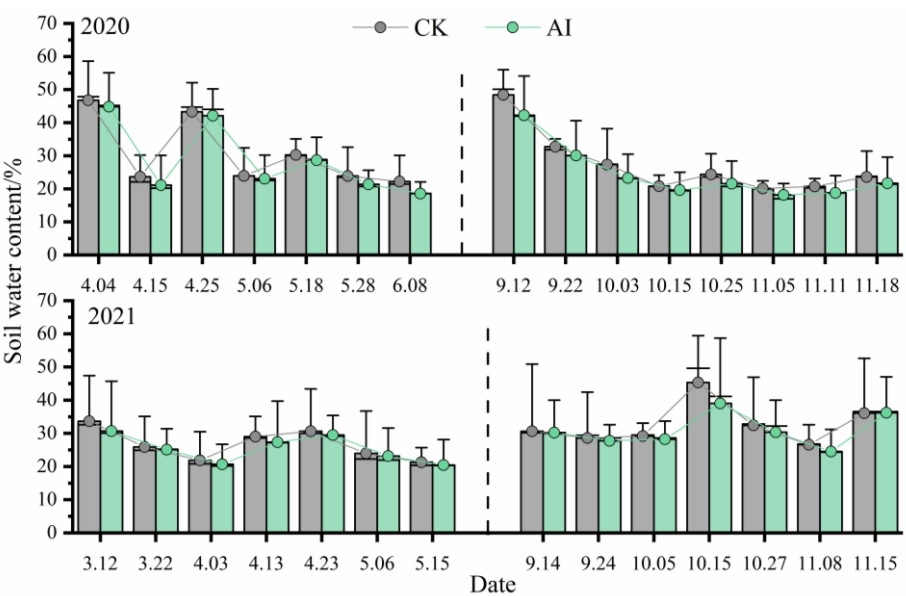

**Figure 5.** The dynamic change curve of soil water content under different treatments (AI, aerated irrigation; CK, unaerated subsurface drip irrigation).

### 3.1.5. Soil Bacterial Biomass

Soil bacteria account for about 70–90% of the total soil microorganisms and are the main influencing factor on soil respiration [7]; there were basically significant differences ($p < 0.05$) in all measured points of soil bacterial biomass under different treatments (Figure 6), and the bacterial biomass under AI treatment was significantly higher by 33.47–45.09% ($p < 0.05$) compared to the control group CK in the 2-year treatment.

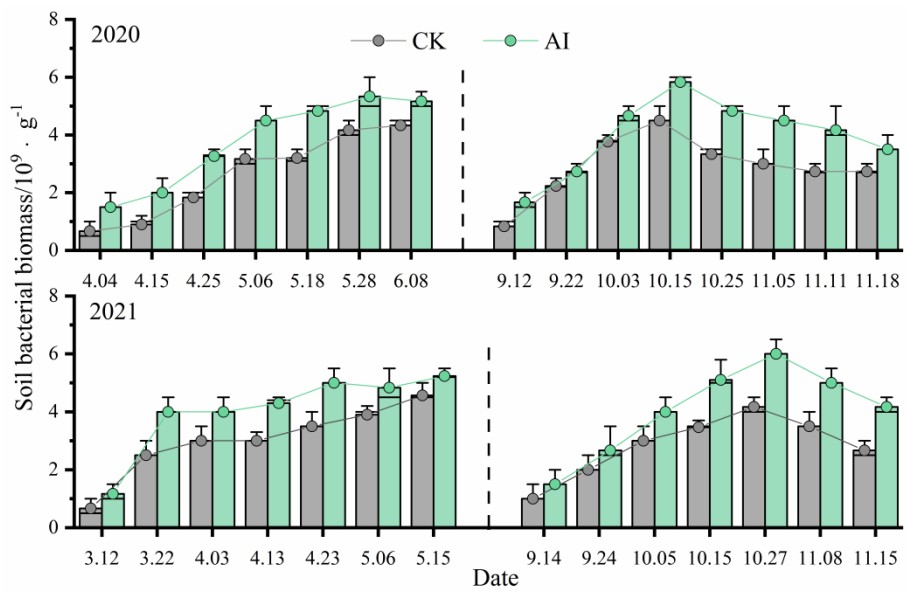

**Figure 6.** The dynamic change curve of soil bacterial biomass under different treatments (AI, aerated irrigation; CK, unaerated subsurface drip irrigation).

### 3.1.6. Root Biomass

Root biomass increased slowly, then rapidly and finally slowly with time after planting (Figure 7). At the seedling stage, the root biomass increased slowly, and the increase was consistent among the treatments, while at the nodulation stage, the root biomass grew faster, and at the maturity stage, the root biomass increased slightly and then decreased, but the range of change was small. The root biomass under AI treatment was higher than the control under different treatments, and the root biomass under AI treatment increased by 15.62–20.12% compared to the control CK ($p < 0.05$).

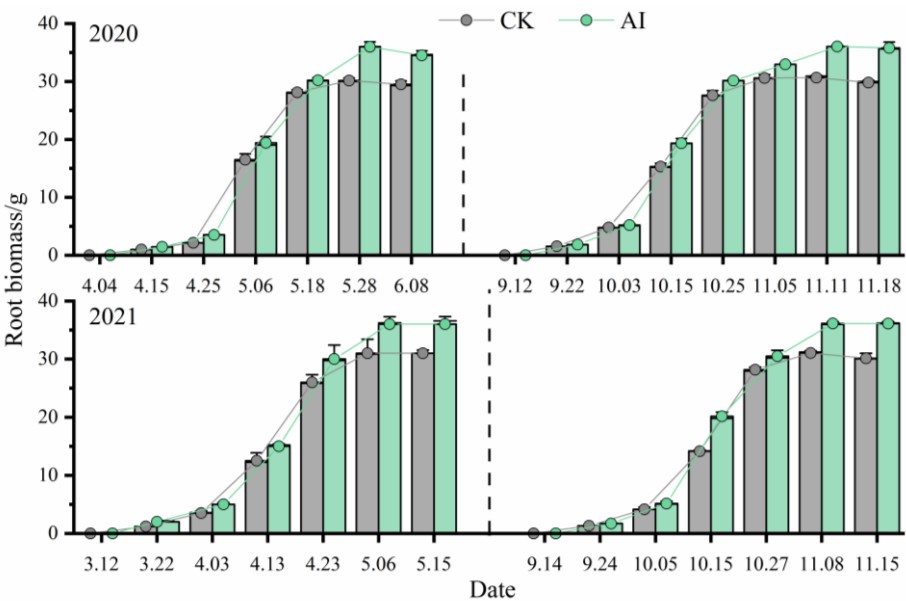

**Figure 7.** Changes in maize root biomass factors under different treatments (AI, aerated irrigation; CK, unaerated subsurface drip irrigation).

### 3.2. Correlation Analysis of Soil Respiration Rate with Soil Environmental Factors under Different Treatments

The results of the experiment were analyzed by segment fitting (excluding the abnormal data in the experiment), and the relationship between the soil respiration rate and each influencing factor was segment fitted by a linear model, nonlinear model and polynomial model, and the best fit equation was screened by the coefficient $R^2$ (Figure 8). Soil temperature was an important environmental factor affecting the soil respiration rate, and there was a negative quadratic polynomial correlation ($p < 0.05$) between the soil respiration rate and the soil respiration rate under both experimental treatments (Figure 8a), with the soil respiration rate peaking at 28.2 °C and then gradually decreasing under the AI treatment, and with a peak at 27.1 °C and then gradually decreasing under the CK treatment. There was a linear positive correlation ($p < 0.05$) between the oxygen content and soil respiration rate (Figure 8b) and a negative cubic polynomial correlation ($p < 0.05$) between the water content and soil respiration rate (Figure 8c). Moreover, in this study, the soil respiration rate gradually increased with increasing soil oxygen content and water content under different treatments. The soil respiration rate was positively correlated with root biomass and bacterial biomass as a power function ($p < 0.05$) (Figure 8d,e), and the soil respiration rate gradually increased with the increase in root biomass and bacterial biomass under different treatments. The correlation analysis between the soil respiration rate and the influencing factors shown in Figure 8f showed that root biomass, bacterial biomass and soil temperature were the important factors affecting the change in soil respiration rate, followed by soil oxygen content, and the effect of soil water content on the soil respiration rate was low in this study.

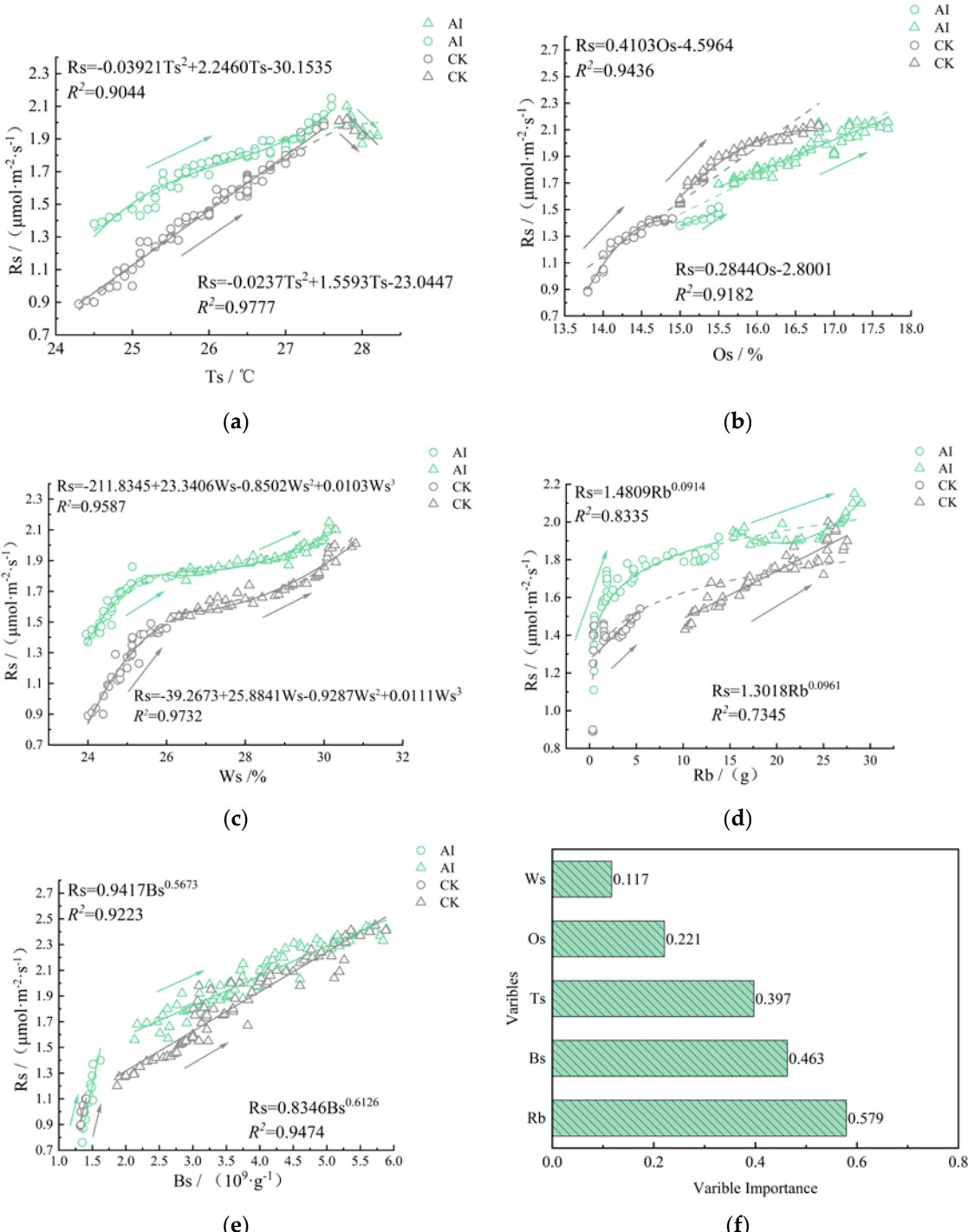

**Figure 8.** Relationships between soil respiration rate and soil temperature (**a**), soil oxygen content (**b**), soil water content (**c**), root biomass (**d**), and soil bacterial biomass (**e**), under AI and CK conditions and importance of variables of the RF model for soil respiration rate (**f**), projection of importance of variables affecting soil respiration rate (VIP) (AI, aerated irrigation; CK, unaerated subsurface drip irrigation).

### 3.3. Soil Respiration Rate Driving Mechanism under Aerated Irrigation

Structural equation modeling (SEM) is a method of exploring causal relationships between variables by fitting data to a model expressing causal assumptions, and such relationships are expressed in the form of causal models, path diagrams, etc. Due to the complexity of the soil respiration rate transformation process, different biochemical

reactions may have synergistic or all the time effects; therefore, the use of SEM can more intuitively describe the driving mechanism of the soil respiration rate under different treatments. The results of SEM analysis show that the fitting index of the model reaches the standard of ideal fitting of the estimated index ($\chi^2 = 1.71$; df = 5; $p = 0.89$; RMSEA < 0.001); therefore, the SEM equation in this study can effectively study the change in the soil respiration rate. The SEM equation in this study can also effectively investigate the driving mechanism of soil respiration rate changes.

Under different treatments, bacterial biomass and root biomass explained 91% of the variation in the soil respiration rate, with root biomass having a greater effect on the variation in the soil respiration rate with a standardized path coefficient of 0.66 and bacterial biomass having a path coefficient of only 0.34. Among them, changes in the soil oxygen content significantly affected bacterial biomass and root biomass, explaining 82% and 59% of the variation in both, with standardized path coefficients of 0.47 and 0.52, respectively. Soil temperature mainly affected root biomass with a path coefficient of $-0.51$, and had relatively little effect on bacterial biomass with a path coefficient of $-0.31$. Soil water content had no significant effect on either bacterial biomass or root biomass.

## 4. Discussion

### 4.1. Changes in Soil Respiration under Different Treatments

Soil respiration is an important pathway for gas exchange between the soil and the atmosphere, a process that consumes $O_2$ and releases $CO_2$ [2,5], mainly from the autotrophic respiration of crop roots and the heterotrophic respiration of soil microorganisms. It has been shown that aerated irrigation treatment can significantly improve the soil oxygen content in the root zone of crops and effectively alleviate the low oxygen stress in the root zone of crops, thus promoting soil respiration. In this study, soil respiration was significantly increased under aerated irrigation compared with the control treatment in both the autumn–winter and spring–summer crop trials. On the one hand, this is because the increase in soil oxygen content provides a good aerobic environment for plant roots and microbial life metabolic activities, and improves root growth and microbial metabolic activity, which in turn enhances the rate of soil respiration. On the other hand, it is because aerating the soil improves soil aeration and promotes gas exchange between the soil and the atmosphere [10], which can ensure smooth soil respiration. The results of Bhattaral, Hou, Zhu Yan, Zang Ming and Li Yuan [6,8–11,26,27] also showed that the soil respiration rate increased from about 12.5% to 20.1% compared to conventional subsurface irrigation.

### 4.2. Changes in Soil Environmental Factors under Different Treatments

In this study, there was basically no significant difference ($p < 0.05$) between all measurement points of soil temperature under different treatments. Soil temperature under AI treatment was higher than that under CK treatment only at some of the measurement points (Figure 3). Soil temperature changes were closely related to climate change and were the main factor influencing soil respiration rate changes, which was also evident from the fact that soil temperature peaks corresponded significantly to soil respiration rate peaks (Figures 2 and 3).

In this study, soil oxygen content was significantly higher ($p < 0.05$) under AI treatment than in the control treatment (Figure 4). Even though soil oxygen consumption was increased by enhanced soil respiration under the aerated treatment, soil oxygen content was still higher than in the control CK throughout the maize reproductive cycle (Figure 4), so the aerated treatment could provide a good oxygen supply environment for the soil in the root zone. This result was confirmed in related studies on aerated irrigation.

The soil water content in this study fluctuated greatly and was mainly influenced by rainfall and irrigation, with rainfall mainly concentrated from September to October. Except for the sudden increase in soil water content due to rainfall, the highest water content was found in all experimental treatments at the maize seedling stage (Figure 5), mainly due to the watering of the bottom water before maize sowing. Soil water content measurement

points under AI treatment were basically lower than those in the control trials (Figure 5), one reason being that under aerated irrigation conditions, soil oxygen content significantly increased, root hypoxia was significantly improved, and plant roots and microbial life activities were vigorous [21], so crop uptake of water and nutrients would be better than the control treatment, and, thus, the soil water content would decrease. Second, because aerated irrigation improved soil aeration, water diffused more evenly in the soil, and avoided the accumulation of water at a certain point in the soil, which can improve the uniformity of soil moisture distribution [10].

Crop root growth was more sensitive to low oxygen stress, and root biomass increased by 15.62~20.12% under aerated irrigation compared with the control, mainly because the improvement in the soil environment provided a good growth environment for crop root growth, met the root's demand for soil oxygen, enhanced the root's uptake of soil water and nutrients, and promoted crop root growth and development (Figure 7). Aerated irrigation improved microbial activity on the basis of improving soil aeration, and the AI treatment significantly increased soil bacterial biomass by 33.47~45.09% compared with the control, and the effect was more significant as the year increased (Figure 8). The seasonal trends of soil bacterial biomass were similar to those of soil respiration, i.e., in the spring and summer crop trials, bacterial biomass generally showed an increasing trend, while in the autumn and winter crop trials, all showed a seasonal trend of increasing and then decreasing (Figures 2 and 8).

*4.3. Correlation Analysis between Soil Respiration and Soil Environmental Factors under Different Treatments*

There are many studies on the relationship between soil temperature and soil respiration, and soil temperature is closely related to the soil respiration rate, affecting almost all aspects of soil respiration, including humus decomposition, root growth and the conduct of various life activities of soil microorganisms [28]. Liu et al. suggested that temperature variation could explain most of the variation in soil respiration rate changes [29]. In this study, soil temperature and the peak soil respiration rate also had a high correspondence, and a segmented fit of soil temperature and the soil respiration rate showed a negative quadratic polynomial correlation between them (Figure 8a), which was inconsistent with studies such as those of Zhu Yan and Niu Wenquan [2,6], mainly due to the different test sites and test climates. Arredondo and other related studies showed that a high temperature can have some inhibitory effects on the soil respiration rate [30].

In this study, the soil oxygen content was significantly and positively correlated ($p < 0.05$) with soil respiration under different treatments (Figure 8b). The results of Arredondo, Hursh, and Anna et al. [30–32] showed that soil respiration was significantly and positively correlated with the soil gas diffusion rate, which increased with the increase in the soil gas diffusion rate. Zhu Yan and Zang Ming et al. also showed that the soil oxygen content was positively correlated with the soil respiration rate, which is consistent with the results of this study.

Soil moisture is also the main controlling factor affecting soil respiration, and by segment fitting analysis, the soil water content was positively correlated with the soil respiration rate three times (Figure 8c). The soil respiration rate did not change significantly when the soil water content was between 24.5% and 30.2%. Hou Mao Mao et al. [33] showed that only when the soil water content was below or above the threshold value was the soil respiration rate significantly correlated with the soil water content, and when the soil water content was between 24.5% and 36.7%, the correlation with the soil respiration rate was weak.

Soil respiration is not only influenced by soil abiotic factors, but more often by biotic factors. The number of plant roots and microorganisms in the soil is an important biotic factor affecting the soil respiration rate. Soil microbial respiration accounts for 40% of soil respiration, root respiration accounts for 40–50% of soil respiration and the rest is $CO_2$ production from the decomposition of apoplastic and organic matter in the soil. Previous

studies have shown that an increase in soil microbial biomass enhances soil respiration, and the soil respiration rate gradually increases with the increase in soil bacterial biomass and root biomass [34]. In this study, root biomass, bacterial biomass and the soil respiration rate were positively correlated as a power function (Figure 8d,e), which is consistent with the findings of previous studies.

Both the random forest model and structural equation model results indicated that changes in bacterial biomass and root biomass had significant effects on changes in the soil respiration rate (Figures 8f and 9), and in the structural equation model, the soil oxygen content and soil temperature had no direct effect on the soil respiration rate, but mainly affected the soil respiration rate indirectly by affecting bacterial biomass and root biomass. Soil bacterial and root biomass play a crucial role in soil carbon release [35]; however, there may be significant redundancy in the functions of soil microorganisms and plant roots, and changes in soil biological properties below a certain threshold may have little effect on soil function [36]; thus, soil microorganisms and plant root mineralization are influenced by soil abiotic factors such as temperature, oxygen content and pH, and active organic matter content factors are strongly regulated, which is consistent with the results of this study. Although aerated irrigation had some effect on raising soil temperature, the effect was not significant, and the change in soil temperature was mainly related to atmospheric temperature (Figure 3) and not greatly related to whether the aerated treatment was applied or not. Therefore, it can be concluded that the soil oxygen content, root biomass and bacterial biomass are the most important environmental and biological factors affecting the soil respiration rate under the effect of aerated irrigation.

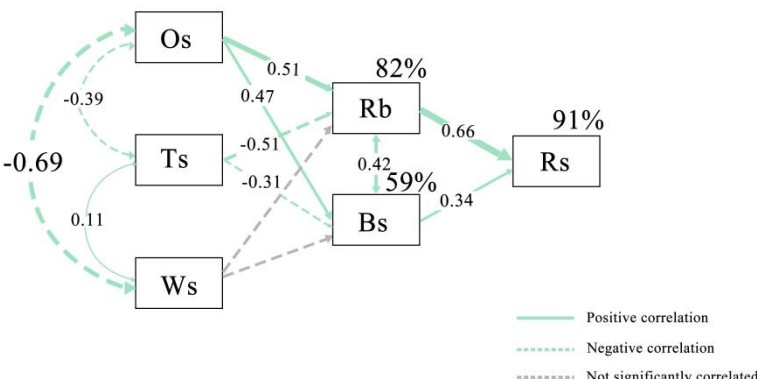

$$\chi^2 = 1.71; \ df = 5; \ p = 0.89; \ RMSEA < 0.001$$

**Figure 9.** Driving mechanism of soil respiration rate change in structural equation model. Note: Rs, soil respiration rate; Ts, soil temperature; Ws, soil water content; Os, soil oxygen content; Bs, soil bacterial biomass; Rb, root biomass.

### 4.4. Recommendation and Limitations

This study not only analyzed the impacts of AI on the soil respiration rate, soil temperature, water content, oxygen content, soil bacterial biomass and root biomass, but also explored the relationships between these parameters. However, our study only examined a subset of the factors influencing the soil micro-environment, and further research on AI should systematically analyze the impacts of AI on other soil parameters, such as soil enzymes, soil organic matter and soil gas (including $CH_4$ and $N_2O$) exchanges. Further research in this field will be carried out in the future.

### 5. Conclusions

(1) This was obtained from a study of four crops in two consecutive years from 2020 to 2021. The soil respiration rate showed a trend of increasing and then decreasing at different fertility stages of maize under different treatments. Compared with the control CK, AI

treatment significantly increased the soil respiration rate by 16.25% to 20.31%, increased soil oxygen content, root biomass and bacterial biomass by 18.07% to 26.12%, 15.62% to 20.12% and 33.47% to 45.09%, respectively, and decreased soil water content by 4.29% to 12.85%, and the soil temperature was not significantly affected by different treatments.

(2) Correlation analysis showed that the soil temperature was negatively correlated with the soil respiration rate using a quadratic polynomial correlation, the soil oxygen content was linearly correlated with the soil respiration rate ($p < 0.05$), root biomass and bacterial biomass were positively correlated with the soil respiration rate by power function, the soil water content was positively correlated with the soil respiration rate by cubic polynomial and the soil respiration rate increased with the soil oxygen content, root biomass and bacterial biomass. The soil respiration rate increased gradually with the increase in soil oxygen content, root biomass and bacterial biomass.

(3) Aerated irrigation technology can drive the increase in the soil respiration rate by changing the soil oxygen content, root biomass and bacterial biomass.

**Author Contributions:** J.C. and F.T. conceived the study and designed the project. J.C. performed the experiment, analyzed the data and drafted the manuscript. F.T. helped to revise the manuscript. All authors have read and agreed to the published version of the manuscript.

**Funding:** This study was funded by the Natural Science Fund Key Project of Heilongjiang Province (ZD2019F002).

**Informed Consent Statement:** Informed consent was obtained from all subjects involved in the study.

**Data Availability Statement:** Not applicable.

**Conflicts of Interest:** The authors declare no conflict of interest.

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
