# Peer review of "PLSR-Based Assessment of Soil Respiration Rate Changes under Aerated Irrigation in Relation to Soil Environmental Factors"

_agriculture, doi:10.3390/agriculture13010068_

Round 1

Reviewer 1 Report

The study is innovative, nicely conducted and reported. My only concern that there were in-fact only two treatments, control and aerated irrigation systems replicated thrice. Also, plot size where the subsurface drips were installed is not mentioned in the MS. 

Line 38 (Abstract): Pl these figures 83×109 to 38 108×109 t of ---

Relate the studied parameters with test crop yield also.

Also, give implications of the study in the conclusion 

Overall, nice work

Author Response

请参阅附件

Reviewer 2 Report

Dear colleagues, I have studied your paper.
I have several reservations, some are just minor things, some are serious.

The introduction could be a bit more comprehensive and less general, it would be advisable to slightly increase the number of cited sources.

The first part of the methodology is well described and contains all the necessary information. Table 1 is perhaps a bit redundant, but that's just a detail. A label would probably be enough, but if you've already created one, leave the table in place.

The discussion is very extensive with a number of literary sources. The conclusion then corresponds to the paper.

Author Response

请参阅附件

Reviewer 3 Report

Reviewer comments

1. In Table1, Soil texture, soil organic matter and soil porosity should also be given. Because, these soil environment factor under aerated irrigation are affected soil respiration.

2. What is the use of soil pH in Table 1 in the research? It is not clear?

3. Aeration irrigation in addition to root biomass, has an effect on shoot biomass.

4. It is recommended to form a multivariable regression between soil respiration and environmental variables.

.

5. Figure 8 is not clear. A clear flow chart should be replaced.

6. Please correct the units in Figure 1.Maximum temperature (mm), Minimum temperature (mm).

7. Methods adequately are not described.

8. English language and style, moderate changes required.

Author Response

请参阅附件。
